# The Importance of Biofilms on Microplastic Particles in Their Sinking Behavior and the Transfer of Invasive Organisms between Ecosystems

Christine C. Gaylarde [1,2], Marcelo P. de Almeida [3,4], Charles V. Neves [4], José Antônio Baptista Neto [3,4] and Estefan M. da Fonseca [2,3,4,*]

1 Department of Microbiology and Plant Biology, Oklahoma University, 770 Van Vleet Oval, Norman, OK 73019, USA
2 Programa de Pós-Graduação em Administração/Mestrado—PPGAd. Rua Mário Santos Braga, S/N—4° Andar—Prédio 1 CEP:24.020-140—Campus do Valonguinho—Centro, Niterói 24210-340, RJ, Brazil
3 Programa de Pós-Graduação em Dinâmica dos Oceanos e da Terra Av. Gen. Milton Tavares de Souza s.n CEP:24.020-140—Campus do Gragoatá—Centro, Niterói 24210-340, RJ, Brazil
4 Aequor-Laboratório de Inteligência Ambiental. R. Joaquim Eugênio dos Santos, 408-Eldorado, Maricá 24901-040, RJ, Brazil
* Correspondence: oceano25@hotmail.com

**Abstract:** Although plastic is ubiquitous in marine systems, our current knowledge of its transport is limited. Recent studies suggest size-selective removal of small plastic particles (<5 mm) from the ocean surface as a result of the formation of a biofilm (the "plastisphere") on the microplastic particle (MP) surface. This localized microenvironment can isolate the microcosm from the adjacent aqueous medium, and thus protect component alien species from the surrounding physico-chemical conditions. Apart from resulting in specific conditions for the transfer of alien species through the environment, the plastisphere can impact MP hydrodynamics and cause MPs to move through the water column, initially sinking. The importance of this phenomenon has not been previously considered for these particles. The size-dependent vertical movement of MPs through the water column determines their distribution, which will vary with time of exposure and colonization. Some plastisphere organisms have plastic-degrading activities, which could be harnessed in marine depollution strategies. This article seeks to increase our understanding of the role of biofilms in the biological dynamics and diffusion of plastic microparticles.

**Keywords:** biofouling; hydrodynamics; sediments; plastics biodegradation; plastisphere

## 1. Introduction

Nowadays, plastic and its subproducts are omnipresent in all aquatic environments, from onshore aquatic ecosystems to oceanic environments [1–4], and from intensely occupied centers [5] to remote sites [6,7]. In spite of this, there is little information about the dynamics and mechanisms of diffusion of plastic microparticles from their origins to their deposits around the planet. There is a particular lack of knowledge about underwater environments, since the vast majority of studies focus on transport of floating particles in the surface layers [3,8–10]. It is estimated that surface transport moves thousands of tons of plastic waste [3,4,11], with only 1% being of continental origin [11–13].

The term microplastic is generally used for particles varying from 5 mm to micrometers in size [1–4,14,15]. This "new" class of pollutant includes cosmetics, synthetic fabric fibers [5], industrial raw resins [1,6], paints [16], as well as deteriorated larger plastics [6]. It has been suggested that the quantity of floating microplastic particles at the surface is lower than projected from the calculated rate of fragmentation of buoyant plastic [17]. Palatinus et al. (2019) [18], however, found that the quantities of macro and micro plastic detected at the surface in the middle Adriatic Sea were correlated in channel waters, although

not in the open sea. It is expected, then, that the ocean floor will represent an effective deposition site for heavier microplastics [6]. Although Chubarenko et al. (2016) [19] reported low levels of lighter microplastic particles on the seafloor, low density microplastic was present in high amounts in offshore subtidal sediments [20,21].

A large part of the plastic particles that enter the aquatic environment sink in the water column and are, at least temporarily, deposited on the ocean floor [6,7,10]. However, Palatinus et al. (2019) [18] showed no correlation between quantities of seabed and floating MPs in the middle Adriatic Sea; they explain this by postulating horizontal transport of MPs by sea currents during the sinking and resuspension processes. On the other hand, many plastic residues are very buoyant [11] and are present for a longer time in the superficial aquatic strata. For instance, polyethylene (PE) shows a lower density if compared with water characteristics, representing more than 70% of the global plastic release in the environment [1]. Even so, some lighter density plastics are present on the ocean floor, evidencing existing processes capable of stimulating their deposition. Many authors suggest that sinking of lower density plastics is a result of their increased weight caused by surface biofilms [1,6,19,22–24]. This biofouling consists of the colonization of the surface of the plastic particle by organisms through the secretion of organic matrices, enhanced by the hydrophobic character of plastic [25]. The rapid adsorption of organic compounds produces a so-called "conditioning film" [22,26], allowing the initiation of a biological succession that starts with bacterial colonization and is followed by the attachment of microalgae and potentially by invertebrate groups [1]. The fixation and colonization is directly dependent on the polymer composition and surface area [22], as well as other characteristics such as surface roughness and the physical chemical characteristics of the medium [1,26,27]. Thus, the surface colonization of plastic particles is a result not only of the initial adhesion of organisms, but also the characteristics of the polymer, the environment in which the particle is located and the season of the year; the latter will ultimately determine characteristics such as temperature, sunshine and resulting primary productivity [1,2,22,26,28,29].

Biofouling by biota leads to heavier particles, which will sink more rapidly [22,24]. Nevertheless, the rate at which this process takes place on plastic microparticles is uncertain [2]; the process of physical transfer of particles between surface and deeper water is still unclear. In this context, surface biofouling is considered an effective mechanism to stimulate buoyant microplastic deposition, at least theoretically [19]. However, although it occurs on many kinds and shapes of plastic [2], the small size of MPs limits the deposition resulting from biofouling.

## 2. Biofouling (Biofilm Formation)

Biofouling consists of the colonization of solid objects by organisms in aquatic sites. Microbial adhesion to and subsequent colonization of MPs in the aqueous environment is rapid, beginning within minutes [30] (Figure 1).

Biofilms include microbial cells, bacteria, algae, protozoans, and fungi, covered by an extracellular matrix. This represents the largest part of the biofilm and is composed principally of exopolysaccharides, with internal channels for the circulation of water, enzymes, nutrients and waste. The community forming the biofilm is made up of various microbial species found on local natural substrates [14,25,31–34]. Environmental aspects directly influence colonization and the ecological balance [2] (Figure 2). The substrate influences surface colonization and ecological succession through the availability of toxic constituents and additives of the plastic matrix. The plastic surface thus governs species selectivity by the impact of its functional properties on the cells' metabolism [35]. For instance, there is an increase in metabolic rate and change in biogeochemical activity in plastic-associated biofilms compared to the local microbiota; the oxygen concentration increases and the expression of genes responsible for secretion, chemotaxis, cell-cell interactions, and nitrogen fixation are modified [31,36].

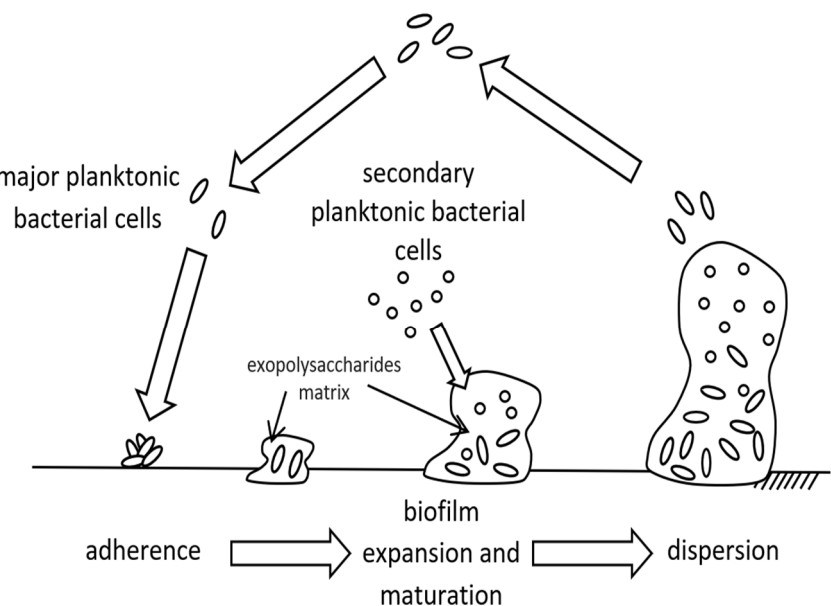

**Figure 1.** Subsequent stages of biofilm formation and microplastic surface colonization.

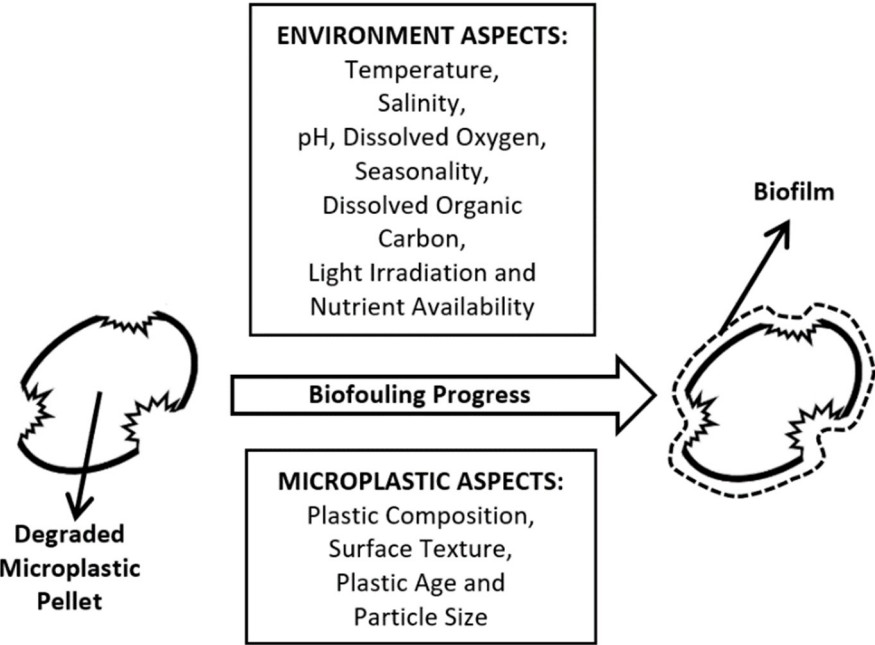

**Figure 2.** Site and microplastic influential aspects of biofilm formation.

In addition to the ecological processes occurring on the MP surfaces, biofouling also impacts characteristics of the plastic material, such as hydrophobicity and buoyancy, since it modifies the volume:density ratio [12,37,38]. With the resulting increase in relative density overcoming the density of the liquid in which the object is immersed, the particle tends to move downward [12,37,38]. Through the increasing depth and resulting rise in pressure, the particle tends to enter into density equilibrium with the aqueous medium and potentially remain in suspension [22]. Ye and Andrady (1991) [22] suggested that the equilibrium depth may coincide with the pycno- and thermo-cline vertical zone. On the other hand, some particles do not come into equilibrium with the density of the medium, causing them to sink to the bottom. Many researchers have recorded MPs deposited on the ocean floor, although the deposition processes remain uncertain [38–40].

The microplastic surface colonization process (Figure 3) can take many days. The formation of the primary, microbial, biofilm, the so-called "plastisphere" [25], influences the biochemical dynamic of MPs in aquatic ecosystems. The plastisphere represents the "interface" between solid and aquatic media and, as a consequence, controls the interactions between the plastic surface and the aquatic environment [41]. Algal succession generally begins later, although diatoms may sometimes [42], though not always [43], be found mixed with the primary microbial biofilm. Other algae may become more apparent after weeks [12]. Ye and Andrady (1991) [22] described the formation of a "transparent slimy biofilm on the surface" after some days. MPs may, therefore, begin to sink after some weeks, under the influence of the attached biota, depending on particular characteristics of the particle, such as size, composition, shape, and roughness, as well as the environmental conditions [44]. Ye and Andrady (1991) [22] described plastics sinking over 7 weeks.

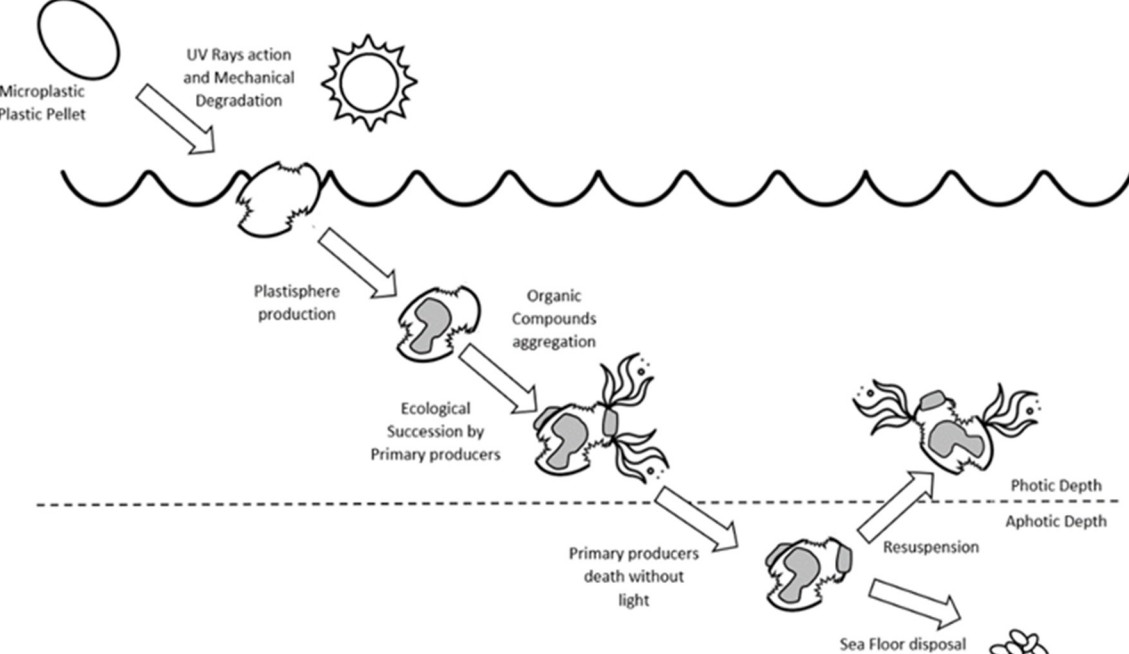

**Figure 3.** The various stages of the biofouling process.

On the other hand, a decrease in biofilm mass, called defouling, can also occur, resulting from light limitation at deeper layers, grazing, or dissolution of carbonates in acid waters [44]. This may allow particles to begin to rise again through the water. It can be followed by a new colonization occurring under submerged conditions, though with different algal species and at slower rates [12]. Rummel et al. (2017) [45] suggested that evaluating the impacts of biofilms on the vertical transport of MPs should be a priority to help us to understand the fate and effects of MPs in aquatic environments.

The speed of biofouling is controlled, as well as by the metabolic activities of the adhering organisms, by the shape and texture of the MPs, the physicochemical characteristics of the environment, such as temperature, radiation and nutrient availability, and water column aspects. Microplastic particles may thus occupy different depths at different times in the water column, depending on the degree of biofouling. As a consequence of biofilm production by the biota, suspended MPs can travel vertically through the water column or remain at the same depth, diffusing horizontally. The production of a biological matrix over the particle surface can also impact the plastic aging processes, for example, protecting the particle from UV [45].

The MP biofilm can change its crystallinity, stiffness, and maximum compression properties. During 2 weeks of exposure to a bacterioplankton assemblage from the Baltic Sea, PE MPs showed an increase in crystallinity (Xc > 82%), polypropylene (PP) MPs

showed a decrease in stiffness by an average of 35 N mm$^{-1}$ and polystyrene (PS) MPs showed an increase in maximum compression ($\varepsilon$max), with the exposed PS being more resistant to breaking down. Both PP and PS MPs showed significant changes in surface chemistry detected by ATR-FTIR [46]. These physicochemical changes could be due to the biodegradation of additives in the plastics. Such changes can lead to a decrease in MP hydrophobicity, which may decrease the sorption of organic contaminants [47]. On the other hand, any increase in negative surface charge following biofilm formation or polymer degradation can enhance sorption processes or increase the absorption of organic contaminants from the seawater [48].

Ultimately, the association of biofouling and dissolved organic matter attached to the microplastic surface impacts the fate and diffusion of MPs in aquatic environments. The varied chemical and mechanical changes produced by biofouling in the presence of marine sediment have been shown to cause an increase in particle density by a combination of biofouling and deposition of organics [49]. This increased density leads to sinking of the MPs. Thus aquatic environments with high levels of dissolved organic matter tend to have higher concentrations of MPs in their sediments [49].

### 3. The Plastisphere

The particular characteristics of the plastic matrix, such as its floating ability and hydrophobicity, have created a new unique substratum for microbial colonization [25,50,51]. The new micro-niche thus created becomes occupied by a specific biofilm called the plastisphere [25,42,52–55].

The total mass of the plastisphere in the oceans cannot be neglected, representing about 0.01–0.2% of the total microbial biomass in their surface waters [42]. However, because of the unknown total amount of plastic discarded in the oceans, the total mass of the plastisphere may be much higher than this [3,42]. Indeed, some authors have described MPs and their associated plastisphere as the eighth continent [52,56,57]. More research on the plastisphere and its importance in biogeochemical cycling and the resulting environmental balance [58] is fundamental.

As a result of the different physicochemical conditions in fresh and saline water, the microbiota in these two ecosystems is distinct, which can impact the structure and evolution of the microbial populations in these environments [53] The microbial ecology of the plastisphere, however, is mainly controlled by the composition of the colonized plastic [59]; MPs work as a filter for microorganisms in the environment.

As hydrophobic organic surfaces with large surface area:volume ratios, MPs readily attract organic matter from the water column, including organic carbon sources and pollutants such as pesticides [60] and hydrocarbons [61]. In addition, many of the chemical compounds added to plastics during their industrial production are toxic to the colonizing microorganisms. These characteristics turn the MP surface into a very complex substratum that is highly selective for colonization by specific microbial species.

Nowadays, thanks to new technologies based on metagenomics, it has been possible to observe the complexity and partially understand the operation of the plastisphere. Reisser et al. (2014) [62] and Dussud et al. (2018) [33] confirmed the influence of certain properties of plastic fragments such as composition, size, degree of degradation, and surface roughness. Amaral-Zettler et al. (2015) [42] noted important differences between the microorganisms colonizing MPs in two different oceans, and between planktonic and sessile cells on MPs in the same environment. Oberbeckmann et al. (2018) [2] and Debroas et al. (2017) [63] showed that the microbial communities present on the surfaces of marine MPs are very different from those in surrounding middle and upper waters or on other particle types (Figure 4). The authors reported greater abundance and richness of colonizing bacterial assemblages on a natural substrate compared with MP communities. This suggests that the modern universal availability of MPs in our oceans not only affects the structure, composition, and functional properties of attached bacteria but also represents a potential

ecological risk as a function of the high stability, pathogenicity, and stress tolerance of the bacterial communities present on the MP surface.

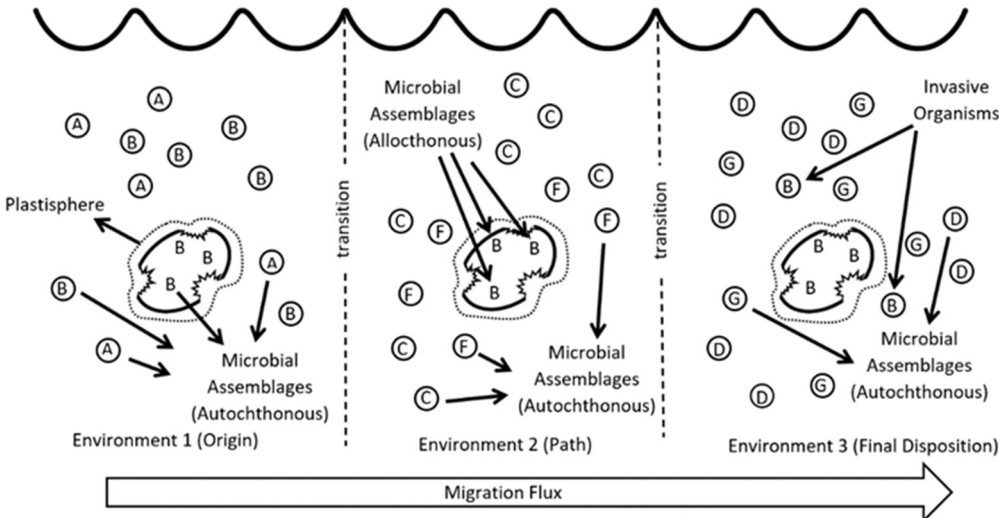

**Figure 4.** Transport of invasive species during microplastic migration along different sites (The varied strains of bacteria are represented by different letters: A, B, C, D, F and G).

Some bacterial groups, such as the phyla Bacteroidetes, Proteobacteria, Cyanobacteria and Firmicutes, are more often found colonizing MPs than other types of particles [25,33]. Certain bacterial taxa, then, seem to be more resistant to the toxic compounds of the plastic matrix, either naturally, or because of ready metabolic adaptation. The latter may be linked to processes such as attachment, degradation or chemotaxis [25,33].

Under the protective impact of the plastisphere, MPs can translocate the local microbiota to other areas, "rafting" microorganisms from their origins to other ecosystems [52,59]. Plastic items produced by humans and discharged into the marine environment as wastes can therefore be responsible for the migration and transportation of allochthonous species in aquatic environments (Figure 4). In this way, it has been suggested, pollution-resistant [64] or antibiotic-resistant [64,65] microbial groups may spread worldwide [52,66].

Human and non-human pathogenic bacteria have been detected in the plastisphere, again indicating the importance of this protective milieu for disease transmission. One of those most commonly reported is the genus *Vibrio*, which contains species pathogenic to humans [67] and to crustaceans [68]. *E. coli* pathotypes have also been detected in marine plastispheres [69]. In addition, micro-algae and cyanobacteria responsible for algal blooms have been implicated in plastisphere-associated transfer [33]. The adherent organisms may be released from the plastisphere when it breaks down because of a change in environmental conditions or through the action of biodegradative organisms within it.

## 4. The Plastisphere Micro-Niche and Biodegradation

According to Ward et al. (2022) [70], there are significant changes in colony formation during the first weeks of plastisphere production, revealing a complex ecological succession during the period of colonization of the micro-niche. Erni-Cassola et al. (2020) [71] reported that bacteria capable of using hydrocarbons as a carbon source play an important role in the initial stages of the process of colonization of the plastic surface. Similarly, Teughels et al. (2006) [72] and Rummel et al. (2017) [45] believe that the first stages of ecological succession and resulting colonization are dominated by species more adapted to more hostile environments, pioneer substrate-specific taxa capable of degrading plastics, later replaced by more generalist biofilm component species [41]. Initially, bacteria and diatoms are the major biofilm components, but other organisms, such as microalgae, fungi and heterotrophic protists (flagellates and ciliates), also populate these micro-niches. They may bring other degradative activities to the plastisphere. Degradation of plastics

in the marine environment has, however, been less studied than in freshwater or soil, and degradation rates are practically unknown [73]. Goudriaan et al. (2023) [74] discuss the problems and deficiencies of studies on biodegradation of plastics in the marine environment. Unambiguous proof of microbial degradation and quantification of the normally low degradation rates are two problematic areas. There are, however, numerous studies of biodegradation in other environments [75–82].

During biofilm maturation and microbial succession, biological transformations occur in parallel with physical and chemical changes that include degradation and oxidation of the polymer itself by microbiota living on the plastic particle surface in an ecologically complex multilayer micro ecosystem [46]. Microorganisms may be both stimulated and inhibited within the highly variable physicochemical microclimate of the MP surface, depending on the additives and contaminants present. The plastic biodegradation process depends on many variables, such as polymer composition and resulting molecular weight, particle surface physical characteristics and environmental parameters [83–85]. The process has been evaluated by monitoring a varied group of parameters. These are substrate weight loss, changes in mechanical properties and/or chemical structure of the polymer, and the percentage of carbon dioxide released. The initial tests of microbiological biodegradation sought to prove that microbial activity would result in physical changes in the polymer matrix, such as mechanical strength, degree of crystallinity and water absorption [86,87]. The various plastic biodegradation processes are directly related to the compositional particularities of each polymer, just as the active sites of enzymes are particular to their specific substrate configurations. The main polymeric compounds can be divided into three groups: polymers whose basic molecule is formed by linear carbon chains (polyethylene—PE, polypropylene—PP, polystyrene—PS, and polyvinyl chloride—PVC); polymers with ester-linked backbones and side chains (polyethylene terephthalate—PET, and polyurethane—PU); and polymers with hetero/carbamate(urethane) linkages (polyurethanes—PUs) (Figure 5).

**Figure 5.** Structures of major commercial synthetic polymers.

## 5. Linear Carbon Chain Axis Polymers

PE, PVC, PS and PP represent linear carbon chain-based axis molecules. They are widely used in industry [88]. For instance, polyethylene represents the most abundant plastic waste discarded in landfills in the form of plastic bags (69.13%) [89]. Polystyrene (PS), on the other hand, has been the most abundant plastic produced around the globe and is largely used in packaging materials produced for food and disposable dishware [88].

The natural decomposition of linear carbon chain axis polymers begins with the incidence of UV-radiation and oxidation reactions, decreasing their molecular weight, making them amenable to biodegradation. In the specific case of PE, the first biodegradation

steps (UV and oxidative enzyme action) produce carbonyl-groups in the structure. Microorganisms then promote secondary matrix fragmentation, producing metabolites which can be assimilated by bacterial and fungal species. Montazer et al. (2019) [90] regard bacterial species such as *Pseudomonas putida*, *Acinetobacter pittii*, and *Micrococcus luteus* as species that use PE as a source of biomass. PS molecules, on the other hand, are more stable and this, combined with their strong hydrophobic character, results in higher resistance against biodegradation [91,92]. Their carbon–carbon axis structure imbues them with high resistance to enzymatic action; nevertheless, plastic-degrading enzymes can be found in microorganisms from several sources [89]. Some microorganisms, such as *P. aeruginosa* [93], and *Curvularia* species [94], have been observed to degrade PS. The rate of PS degradation can be improved by adding polymer-starch blends, which stimulate molecular transformations [95–97].

Until now, PP is the most widely used linear hydrocarbon polymer among the synthetic polymers. Despite that, there are only a few studies on PP biodegradation. For instance, bacteria of the genera *Pseudomonas* and *Vibrio*, and the fungus *Aspergillus niger*, have been reported to degrade PP [88,98]. However, most studies have been carried out using pretreated PP. The pretreatment techniques involve gamma-irradiation [99], UV-irradiation [100,101], or thermal treatment [102]; these can reduce hydrophobicity or introduce more degradable groups such as C=O or –OH. The latter groups may be formed during degradation of the polymer, along with a decrease in viscosity [99]. UV treatment has been shown to allow the degradation of PP by *Bacillus flexus* [103]. Biodegradation of PP has also been improved by blending it with carbohydrates, starch or cellulose, similar to that reported for PE, PS and PU. The blends facilitate adhesion of the microorganisms to the polymer surface and act as co-metabolites [98,101,102,104,105]. Biodegradation of polycaprolactone (PCL)-blended PP has also been demonstrated using lipase; this group of enzymes is known to degrade the ester linkages of PCL [106].

Finally, PVC does not show a hydrolysable ester bond, making its degradation more difficult. Some authors, based only on morphological and physicochemical changes observation, suggested the possibility of PVC biodegradation by some bacterial taxa (i.e., *Pseudomonas*, *Mycobacterium*, *Bacillus*, and *Acinetobacter*) [107–110].

## 6. Polymers with Ester-Linked Backbones and Side Chains

PET is readily partially biodegradable because of the presence in its structure of hydrolysable polyester bonds. Although several microbial transformations of this plastic had been identified in earlier years [111], it was not until 2016 that Yoshida et al. (2016) [108] isolated an enzyme complex (designated PET-ase) from the bacterium *Ideonella sakaiensis* derived from a bottle-recycling facility. The rate of PET degradation by this enzyme complex was, however, too slow for it to be of practical use and more recent studies have worked on genetically manipulating the genes involved [112–115], or, most recently, on protein engineering [116–118]. The latter has achieved faster rates, more stable enzymes and complete degradation of PET under mild conditions. One recent improvement has been the production of a mirror-image version of PET-ase that is not, itself, biodegraded in natural environments [119]; this biostable enzyme should have longer-acting activity in open ecosystems. Several groups around the world are continuing to work on microbial enzymes that can degrade plastics.

PUs are the sixth most used polymers in the world. They are specifically designed to achieve long-term durability and resistance to biodegradation; they are, however, susceptible to slow biodegradation under specific conditions [120]. This biostability means that most PU waste is currently disposed of in landfill [121] and perhaps the major research effort is devoted to developing more biodegradable types of PU [122–126], rather than PU biodegradation systems.

Although polymer structure is undoubtedly linked to biodegradability, Miao et al. (2023) [127], studying the colonization of various types of "biodegradable" and "non-biodegradable" plastics in freshwater ecosystems, determined that the factors

influencing the composition of the bacterial and fungal surface communities were in the order location > time > plastic type. Our understanding of the initial phases of the ecological succession on MPs and succeeding biodegradation is thus still very limited [128,129].

There are several published reviews on the biodegradation of polymers by microorganisms [130–133] but there is a lack of real information about the relationships between polymer characteristics and microbial communities populating MP surfaces [134]. There have been publications on changes in the physicochemical properties of polymers promoted by microbial action, including degree of crystallinity [135], weight loss [136], hydrophobicity [60], molecular density [135], morphology [25,62,137,138], and surface reactive compounds [136]. Several researchers have reported hydrocarbon-degrading species colonizing plastic particles in seawater [34,62,134] and a recent review identified *Pseudomonas* and *Bacillus* as the genera most frequently identified as plastic-degrading species in the literature [132]. This could be because these two genera are extremely common in many environments and are readily isolated and identified.

Table 1 contains the names of some of the bacterial and fungal species that have been suggested as potential bio-degraders of plastics in the marine environment. Further groups can be found, for example, in the tables in the review article of Rogers et al. (2020) [139]. Based on the available information, plastic degrading microorganisms have been considered as a potential treatment to ameliorate the large amounts of plastic waste in oceans around the world [140,141] and Gambarini et al. (2022) [142] recently produced a database of microorganisms and (principally) proteins that are linked to biodegradation of natural and synthetic plastics, with this use in mind.

**Table 1.** Putative plastic-degrading microorganisms detected in the marine plastisphere.

| Genus/Species | Type of Plastic | Geographic Location | Comments | Reference(s) |
|---|---|---|---|---|
| *Alcanivorax borkumensis* | PE | Mediterranean Sea | 5–27 m depth | [143] |
| *Alteromonas* | PU | San Diego, USA | Pelagic seawater and seawater tanks | [144] |
| *Arenibacter* | PE | Mediterranean Sea | 5–27 m depth | [145] |
| *Bacillus* spp. | PE | India | Pelagic water | [144,146–148] |
| | PE | Worldwide | Marine waters | [149] |
| | PS | Arabian Sea | Deep sea | [150] |
| *Brevibacillus borstelensis* | PE | India | Seawater | [148] |
| *Erythrobacter* | PS, PE | Baltic Sea | Cold seawater | [151] |
| | PU | | Pelagic seawater and seawater tanks | [144] |
| *Halomonas* sp. | PE | Marine environment | In vitro tests | [152] |
| *Kocuria palustris* | PE | Arabian Sea | Pelagic water | [145] |
| *Marinobacter* | PE | Mediterranean Sea | 5–27 m depth | [143] |
| | PU | San Diego, USA | Pelagic seawater and seawater tanks | [144] |

**Table 1.** *Cont.*

| Genus/Species | Type of Plastic | Geographic Location | Comments | Reference(s) |
|---|---|---|---|---|
| *Marinomonas* sp. | PLA | Mediterranean Sea | Sediment and water | [153] |
| *Pseudomonas* spp. | PE | Tamil Nadu, India | Coast | [147] |
| | PU | San Diego, USA | Pelagic seawater and seawater tanks | [144] |
| | PVC | India | Coastal seawater | [152] |
| | PCL | Japanese coast | Halotolerant strain | [154] |
| *Rhodococcus ruber* | PE | Israel | Laboratory isolate (soil in seawater) | [155] |
| | PET | Antarctic Ross Sea | Cold-adapted | [155] |
| | PS | Zhangzhou, China | Marine mangrove ecosystem | [156] |
| *Streptomyces* sp. | PE | Galway Bay, Ireland | Isolated from marine sponge | [157] |
| | PHA | Galway Bay, Ireland | Isolated from marine sponge | [157] |
| | PCL | Japan | Beach | [154] |
| *Thalassospira* | PU | San Diego, USA | Pelagic seawater and seawater tanks | [144] |
| *Thioclava* sp. | PET | Worldwide | Marine waters | [149] |
| *Alternaria* sp. | PE | Qingdao, China. | Huiquan bay | [158] |
| *Aspergillus* sp. | PHB | Bay of Bengal | Deep sea isolate | [159] |
| | PE | India | Coastal sediment | [147] |
| *Cladosporium* | PU | San Diego, USA | Pelagic seawater and seawater tanks | [144] |
| *Clonostachys rosea* | PCL | Arctic regions | Cold seawater | [134] |
| *Penicillium* sp. | PU | San Diego, USA | Pelagic seawater and seawater tanks | [144] |
| *Saccharomonospora viridis AHK19* | PE | Laboratory culture | Thermophilic strain | [160] |
| *Trichoderma* sp. | PCL | Arctic regions | Cold seawater | [134] |
| *Zalerion maritimum* | PE | Portugal | Seawater | [161] |

Abbreviations: PCL polycaprolactone, PE polyethylene, PET polyethylene terephthalate, PHA polyhydroxy-alkanoate, PLA polylactic acid, PS polystyrene, PU polyurethane, PVC polyvinyl chloride.

Although increasing the biodegradation of MPs by microorganisms may seem an obvious treatment for the excessive quantities of plastic particles in the world's oceans, a complete life cycle assessment of the process must be considered. It has been hypothesized that the carbon released from plastic contaminants in the seas potentially impacts natural biogeochemical cycles resulting in an imbalance of the microbial ecology in marine ecosystems [159]. The search for plastic-degrading microorganisms as a depollution strategy may have unexpected long-term and drastic environmental side effects.

## 7. Conclusions and Future Perspectives

Particle size is a basic factor in the dynamics of diffusion of solid particles, which in marine environments can remain suspended in the water column or deposit on the ocean floor. On the other hand, the attraction of organic matter and the consequent potential for biofouling can directly influence the density of the particle, directly impacting its trajectory. In the case of plastic microparticles, if on the one hand the process is no different, on the other it is much more complex. In addition to having hydrophobic characteristics, unlike mineralogical grains, polymers have an extremely variable composition, allowing for greater surface reactivity. Thus, they have greater versatility, attracting a greater variety of ions from the water.

Such changes in surface composition result in a greater range of degradation, a phenomenon referred to as aging, generating varied physical characteristics in bodies of the same composition and size. Differences in surface roughness will generate specific colonization environments, making it even more difficult to predict the trajectory of MPs in the marine environment.

Environmental conditions, as well as microbial colonization, also directly affect the circulation of MPs. From the moment that living beings are directly linked to environmental characteristics and their variations as a result of seasonality, the local ecology itself begins to determine the trajectory of particles in the water body. In more productive environments, such as the tropics, the evolution of ecological succession is different from that existing in temperate environments. Thus the same plastic microparticle will present different trajectories in different ecological environments, regardless of the local hydrodynamics and granulometry.

Biodegradation of the plastic matrix also shows large variations resulting from the complex composition of polymers. As the biodiversity of the surface micro-niche varies with the chemical composition of the plastic, different plastics carry different communities and their activity will vary according to the environment. Thus, decomposition in tropical sites should be more intense, as a result of the higher incidence of ultraviolet rays and higher temperatures acting together with the more intense biological action typical of environments where the metabolism and resulting primary productivity are greater. A greater knowledge of plastic-degrading organisms, together, perhaps, with genetic manipulation of those already isolated to increase their effectiveness, could lead to a viable method for dealing with the microplastic pollution of our oceans. At present, however, a more productive approach seems to be limiting the disposal, and hence availability to the Earth's aquatic systems, of poorly-degradable plastics.

Finally, the great and complex migration ability of plastic microparticles also has a direct effect on the diffusion of allochthonous species. From the moment that the process of colonization by microorganisms is established on the particles, biofilms are formed, protecting the surface from the external physicochemical conditions of the environment and allowing the survival of the adhering species, which can include macro, as well as micro, organisms. Thus, the transport of invasive species to new ecological niches is enhanced and this can result in undesirable, or even dangerous, changes to local populations and ecosystems.

Further observational studies in different marine environments, controlled trials on factors affecting microplastics transport, degradation and environmental effects and, not least, education of the public, will be necessary in the fight against microplastics pollution of our seas.

**Author Contributions:** C.C.G. and E.M.d.F.: conception, development of the theory, discussion and contribution to final manuscript; J.A.B.N., C.V.N. and M.P.d.A.: discussion and contribution to final manuscript. All authors were involved in the literature searches. All authors have read and agreed to the published version of the manuscript.

**Funding:** The research for this article was funded by Maricá Development Company–CODEMAR and by Conselho Nacional de Desenvolvimento Científico e Tecnológico (CNPq).

**Institutional Review Board Statement:** Not applicable.

**Informed Consent Statement:** Not applicable.

**Data Availability Statement:** No data available.

**Acknowledgments:** The authors are grateful to the Municipality of Maricá for infrastructure and administrative support.

**Conflicts of Interest:** The authors declare no conflict of interest.

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
