# Peer review of "The Importance of Biofilms on Microplastic Particles in Their Sinking Behavior and the Transfer of Invasive Organisms between Ecosystems"

_2673-8023, doi:10.3390/micro3010022_

Round 1

Reviewer 1 Report

This is an interesting and informative overview of a poorly-understood microbial niche in the ocean associated with the surfaces of microplastics.  While the article is generally well written, some of the sentence constructions are unnecessarily verbose (i.e. "on the one hand...on the other hand..." could be a simple assertion followed by "in contrast"). But overall I think readers interested in biodegradation and environmental transport of microplastics will find this review useful.

I have a few minor comments that should be addressed before this article is accepted, as follows:

Lines 73 - 76: This sentence needs to be re-worded, as it is literally false on its face. The characteristics of the polymer, the environment and season cannot determine characteristics such as temperature, sunshine and primary productivity - just the opposite in fact.

Figure Legends: The figure legends should contain enough text to allow readers to interpret a figure without reference to the main body of the text. Only the figure 5 legend meets this standard, while the figure 4 legend give no guidance as to the meaning of the encircled letters in the figure.

Line 166: This sentence needs to be supported by at least one reference.

Line 180: Delete "...totally..." here. Marine and freshwater ecosystems are indeed distinct, but not totally (e.g., both are aquatic).

Line 323: Delete "...and so are a challenge to environmentalists;" this clause is irrelevant.

Lines 382 and 393: Tropical marine waters are hardly the most productive in terms of primary productivity. Subarctic and temperate waters, especially when upwelling currents are present, are considerably more productive than tropical marine waters.

Line 396: "..., but currently a more viable solution seems to be limiting the disposal,...", I have no idea what you're trying to convey here. How can "a more viable solution" be limiting the disposal?

Author Response

Comments and Suggestions for Authors (RESPONDING LETTER)

This is an interesting and informative overview of a poorly-understood microbial niche in the ocean associated with the surfaces of microplastics.  While the article is generally well written, some of the sentence constructions are unnecessarily verbose (i.e. "on the one hand...on the other hand..." could be a simple assertion followed by "in contrast"). But overall I think readers interested in biodegradation and environmental transport of microplastics will find this review useful.

I have a few minor comments that should be addressed before this article is accepted, as follows:

ANSWER:

The authors are very grateful for the contributions

REQUEST:

Lines 73 - 76: This sentence needs to be re-worded, as it is literally false on its face. The characteristics of the polymer, the environment and season cannot determine characteristics such as temperature, sunshine and primary productivity - just the opposite in fact.

ANSWER:

Rewritten as

Thus, the surface colonization of plastic particles is a result not only of the initial adhesion of organisms, but also the characteristics of the polymer, the environment in which the particle is located and the season of the year; the latter will ultimately determine characteristics such as temperature, sunshine and resulting primary productivity [1, 2, 22, 26, 28, 29].

REQUEST:

Figure Legends: The figure legends should contain enough text to allow readers to interpret a figure without reference to the main body of the text. Only the figure 5 legend meets this standard, while the figure 4 legend give no guidance as to the meaning of the encircled letters in the figure.

ANSWER:

Subtitles have been modified for the versions below:

Figure 1. Subsequent stages of biofilm formation and microplastic surface colonization.

Figure 2. Site and microplastic influential aspects of biofilm formation.

Figure 3. The various stages of the biofouling process.

Figure 4. Transport of invasive species during microplastic migration along different sites.

REQUEST

Line 166: This sentence needs to be supported by at least one reference.

ANSWER:

The reference below was added:

Wu, P.; Huang, J.; Zheng, Y.; Yang, Y.; Zhang, Y.; He, F.; Chen, H.; Quan, G.; Yan, J.; Li, T.; Gao, B. Environmental occurrences, fate, and impacts of microplastics. Ecotoxicol. Environ. Saf. 2019, 30, 184. doi: 10.1016/j.ecoenv.2019.109612. Epub 2019 Aug 30. PMID: 31476450.

REQUEST:

Line 180: Delete "...totally..." here. Marine and freshwater ecosystems are indeed distinct, but not totally (e.g., both are aquatic).

ANSWER:

Done

REQUEST

Line 323: Delete "...and so are a challenge to environmentalists;" this clause is irrelevant.

ANSWER:

Done

REQUEST:

Lines 382 and 393: Tropical marine waters are hardly the most productive in terms of primary productivity. Subarctic and temperate waters, especially when upwelling currents are present, are considerably more productive than tropical marine waters.

ANSWER:

Upwelling regions, in both tropical and temperate environments, are the most productive. The issue in focus is that upwelling is a hydrodynamic issue and not a location issue. When talking about greater productivity in tropical environments, it is said that greater insolation energy allows greater productivity, which does not happen to the same degree in a temperate climate zone, unless under the influence of upwelling.

For instance, according to Longbord et al. (2021): …“In tropical coastal waters, phytoplankton production rates are generally high as the factors fuelling productivity (e.g. nutrient supply, solar radiation) are relatively stable (Nittrouer et al., 1995), with production rates (e.g. mean of 1850 mg C m−2 d−1 in the Banda and Arafura Seas; Furnas and Carpenter, 2016; Gieskes et al., 1990) in some cases matching productive upwelling regions (i.e. upwelling season mean 1500 mg C m−2 d−1 in the Ria de Vigo; Álvarez-Salgado et al., 2009)

Still, according to Nittrouer et al., 1995, …“tropical waters generally are considered nutrient poor and therefore should have lower productivity. But tropical coastal waters are productive zones due to permanently higher temperatures and sunlight, with phytoplankton production rates in some cases being comparable to the most productive ocean regions (i.e. upwelling areas)...”

References:

Lønborg, Christian, Moritz Müller, Edward C.V. Butler, Shan Jiang, Seng Keat Ooi, Dieu Huong Trinh, Pui Yee Wong, Suryati M. Ali, Chun Cui, Wee Boon Siong, Erik S. Yando, Daniel A. Friess, Judith A. Rosentreter, Bradley D. Eyre, Patrick Martin, Nutrient cycling in tropical and temperate coastal waters: Is latitude making a difference?, Estuarine, Coastal and Shelf Science, Volume 262, 2021, 107571, ISSN 0272-7714, https://doi.org/10.1016/j.ecss.2021.107571.

Nittrouer, C.A.,  Brunskill, G.J. , Figueiredo A.G.  Importance of tropical coastal environments Geo Mar. Lett., 15 (1995), pp. 121-126

REQUEST

Line 396: "..., but currently a more viable solution seems to be limiting the disposal,...", I have no idea what you're trying to convey here. How can "a more viable solution" be limiting the disposal?

ANSWER:

The text has been clarified and rewritten as follows:

A greater knowledge of plastic-degrading organisms, together, perhaps, with genetic manipulation of those already isolated to increase their effectiveness, could lead to a viable method for dealing with the microplastic pollution of our oceans. At present, however, a more productive approach seems to be limiting the disposal, and hence availability to the Earth's aquatic systems, of poorly-degradable plastics.

Reviewer 2 Report

First of all, thank you for your willingness to share your work with the scientific community. I have read the manuscript (2228200) “The importance of biofilms on microplastic particles in their sinking behaviour and the transfer of invasive organisms between ecosystems” carefully.

This review presents an overview on biofilm formation, the plastisphere, its biodegradation and its role in the behaviour of PM in aquatic environments. A review is always a laborious task, and in this case, the authors have made a great effort. Congratulations. I have found the review very interesting, well-constructed and up to date. It would only be necessary to improve some details in order to publish it in Micro.

The general aspect I recommend changing is the bibliographic references. Instead of writing down [1, 2, 3, 4, 5], I suggest [1-5]. Please, check the whole manuscript.

Other punctual aspects are:

In line 42, please change 5mm to 5 mm

In line 44, change (Browne et al., 2011) to the corresponding number [5] and change the reference for paints [21] to paints [16].

In line 128 (Figure 2), change pH. to pH,

In line 155, please change 35 N/ mm to 35 N mm-1

In line 176, I recommend placing reference numbers in order [52, 56, 57]

In line 209, please replace references with the corresponding number [25, 33]

In line 228, I suggest dividing this section. At this point, maintain “The plastisphere microniche” and, in line 328, create a new section entitled “Biodegradation”: In this way, the manuscript is much easier to read. I also suggest putting sections “Linear carbon chain axis polymers” and “Polymers with ester-linked backbones and side chains” as subsections of “The plastisphere microniche”.

In line 268, I recommend following the same order in Figure 5 as in the text, i.e. PE, PP, PS, PVC, PET and PU.

In line 272, I recommend replacing reference 88 (Plastics Europe 2017) with the latest

one Plastics Europe 2022

In line 275: the use of disposable dishware made of plastic is forbidden nowadays, so I suggest putting this sentence in the past time. For instance: for ages, PS was used in disposable dishware.

In lines 292-293: I suggest adding reference 88

In the section “Linear carbon chain axis polymers”, I miss a paragraph about PVC. Please add.

In the section “Polymers with ester-linked backbones and side chains”, I recommend deleting the name of subsections (lines 307-PET- and 321-Pus-).

In Table 1: I suggest getting more space to the “Type of plastic” column to improve the aspect of the table.

Author Response

Comments and Suggestions for Authors (RESPONDIG LETTER)

First of all, thank you for your willingness to share your work with the scientific community. I have read the manuscript (2228200) “The importance of biofilms on microplastic particles in their sinking behaviour and the transfer of invasive organisms between ecosystems” carefully.

This review presents an overview on biofilm formation, the plastisphere, its biodegradation and its role in the behaviour of PM in aquatic environments. A review is always a laborious task, and in this case, the authors have made a great effort. Congratulations. I have found the review very interesting, well-constructed and up to date. It would only be necessary to improve some details in order to publish it in Micro.

ANSWER:

The authors are very grateful for the contributions

REQUEST

The general aspect I recommend changing is the bibliographic references. Instead of writing down [1, 2, 3, 4, 5], I suggest [1-5]. Please, check the whole manuscript.

ANSWER:

Unfortunately, the editors requested this format.

Other punctual aspects are:

REQUEST

In line 42, please change 5mm to 5 mm

ANSWER:

Done

REQUEST

In line 44, change (Browne et al., 2011) to the corresponding number [5] and change the reference for paints [21] to paints [16].

ANSWER:

Done

REQUEST

In line 128 (Figure 2), change pH. to pH,

ANSWER:

Done

REQUEST

In line 155, please change 35 N/ mm to 35 N mm-1

ANSWER:

Done

REQUEST

In line 176, I recommend placing reference numbers in order [52, 56, 57]

ANSWER:

Done

REQUEST

In line 209, please replace references with the corresponding number [25, 33]

ANSWER:

Done

REQUEST

In line 228, I suggest dividing this section. At this point, maintain “The plastisphere microniche” and, in line 328, create a new section entitled “Biodegradation”: In this way, the manuscript is much easier to read. I also suggest putting sections “Linear carbon chain axis polymers” and “Polymers with ester-linked backbones and side chains” as subsections of “The plastisphere microniche”.

ANSWER:

Unfortunately, making such a change will change all the formatting that the journal has already performed on the article, generating a structural error in the progress of the text.

REQUEST

In line 268, I recommend following the same order in Figure 5 as in the text, i.e. PE, PP, PS, PVC, PET and PU.

ANSWER:

Done

REQUEST

In line 272, I recommend replacing reference 88 (Plastics Europe 2017) with the latest one Plastics Europe 2022

ANSWER:

Done

REQUEST

In line 275: the use of disposable dishware made of plastic is forbidden nowadays, so I suggest putting this sentence in the past time. For instance: for ages, PS was used in disposable dishware.

ANSWER:

Done

REQUEST

In lines 292-293: I suggest adding reference 88

ANSWER:

Done

REQUEST

In the section “Linear carbon chain axis polymers”, I miss a paragraph about PVC. Please add.

ANSWER:

The text below has been added:

“Finally, PVC don’t show a hydrolysable ester bond, making its degradation more difficult. Some authors, based only on morphological and physicochemical changes observation, suggested the possibility of PVC biodegradation by some bacterial taxa (i.e., Pseudomonas, Mycobacterium, Bacillus, and Acinetobacter) [107, 108, 109, 110].”

REQUEST

In the section “Polymers with ester-linked backbones and side chains”, I recommend deleting the name of subsections (lines 307-PET- and 321-Pus-).

ANSWER:

Done

REQUEST

In Table 1: I suggest getting more space to the “Type of plastic” column to improve the aspect of the table.

ANSWER:

Done